

# Capsazepine prolongation of the duration of lidocaine block of sensory transmission in mice may be mediated by modulation of HCN channel currents

Wenling Zhao[1,2,*], Peng Liang[1,2,*], Jin Liu[1,2], Huan Li[1], Daqing Liao[1], Xiangdong Chen[3], Qian Li[1,2] and Cheng Zhou[1,2]

[1] Laboratory of Anesthesia & Critical Care Medicine, Translational Neuroscience Center, West China Hospital of Sichuan University, Chengdu, Sichuan, China
[2] Department of Anesthesiology, West China Hospital of Sichuan University, Chengdu, Sichuan, China
[3] Department of Anesthesiology, Union Hospital of Tongji Medical College, Huazhong University of Science and Technology, Wuhan, Hubei, China
* These authors contributed equally to this work.

Corresponding authors
Qian Li, hxliqian@foxmail.com
Cheng Zhou, zhouc@163.com

## ABSTRACT

**Background and objectives:** Hyperpolarization-activation cyclic nucleotide-gated (HCN) channels contribute to the effects of lidocaine. Capsazepine (CPZ), a competitive inhibitor of capsaicin of transient receptor potential vanilloid-1 channel, has also been found to inhibit HCN channel currents ($I_h$). This study was designed to investigate whether CPZ could prolong durations of lidocaine in regional anesthesia.

**Methods:** Mouse HCN1 and HCN2 channels were expressed in human embryonic kidney 293 (HEK 293) cells. The effect of CPZ on $I_h$ was measured by whole-cell patch-clamping recording. Sciatic nerve block model in mice was used for the study in vivo. The mice were randomly divided into seven groups, respectively, receiving lidocaine, CPZ, ZD7288 (HCN channel blocker), CPZ + lidocaine, ZD7288 + lidocaine, ZD7288 + CPZ + lidocaine, forskolin (an activator of adenylyl cyclase) + CPZ + lidocaine. Regional anesthetic durations of lidocaine were determined. Voltage-gated sodium channel currents ($I_{Na}$) and $I_h$ were recorded in dorsal root ganglion neurons of mice. The effects of CPZ on $I_{Na}$ and $I_h$ with or without Cyclic adenosine monophosphate (cAMP) were assessed. Isolated mice sciatic nerve was prepared to evaluate the effect of CPZ on the compound action potentials (CAP).

**Results:** Capsazepine non-selectively inhibited transfected mHCN1 and mHCN2 channel currents in HEK 293 cells. In sciatic nerve block in vivo, compared to lidocaine alone, adding CPZ extended the durations of lidocaine for noxious sensory block (35.1 ± 3.3 vs. 20.3 ± 1.7 min), tactile sensory block (25.5 ± 4.4 vs. 20.0 ± 3.7 min), thermal sensory block (39.6 ± 6.6 vs. 26.8 ± 5.5 min), and motor function block (28.6 ± 4.1 vs. 20.9 ± 4.2 min). Duration of thermal sensory block was longer in CPZ + lidocaine group than that of ZD7288 + lidocaine group (39.6 ± 6.6 vs. 33.4 ± 4.5 min). Forskolin reversed the prolongation by CPZ on lidocaine durations. CPZ or ZD7288 alone did not produce typical regional anesthetic effects. Increased intracellular concentration of cAMP reversed the inhibition of CPZ on $I_h$. Although CPZ alone inhibited $I_{Na}$ at the concentration more than 30 μM, it did

not inhibit the CAP amplitudes in isolated sciatic nerves. CPZ dose-dependently enhanced the inhibitory effect of 1% lidocaine on the CAP amplitudes.
**Conclusions:** Capsazepine may prolong durations of lidocaine in peripheral nerve block by modulation of HCN channel currents.

## INTRODUCTION

Although local anesthetics such as lidocaine have been widely used in clinics for decades, there are still many disadvantages for currently used local anesthetics including toxicities and unsatisfied durations (*Mather, 2010*). To prolong the duration of local anesthetics, many adjuvants have been added in clinical regional anesthesia (*Kirksey et al., 2015*). However, the enhancement of most adjuvants are unsatisfied and some of the adjuvants can even induce unacceptable complications (*Lundblad & Lonnqvist, 2016*).

Voltage-gated sodium channel is the principal molecular target for local anesthetics (*Becker & Reed, 2012*). Recent studies have identified that hyperpolarization-activation cyclic nucleotide-gated (HCN) channel might also contribute to the effects of local anesthetics in regional anesthesia (*Meng et al., 2011*; *Zhou et al., 2015*). HCN channel is a non-selective cation channel activated by hyperpolarization of membrane potential (*Benarroch, 2013*). HCN channel is the principal pacemaker for the rhythmical regulator and determinant for resting membrane potential in the nervous system and sinoatrial node for its distinct electrophysiological properties (*Benarroch, 2013*; *Biel et al., 2009*). HCN channel also plays an important role in neuropathic pain (*Cao, Pang & Zhou, 2016*). Our previous study indicates that the duration of lidocaine is significantly shortened in HCN1 knockout mice (*Zhou et al., 2015*). Cyclic adenosine monophosphate (cAMP) is the enhancer for HCN channel currents ($I_h$) (*Wainger et al., 2001*). Increased level of cAMP by activator of adenylyl cyclase (e.g., forskolin) can shorten the durations of lidocaine in vivo (*Kroin et al., 2004*; *Zhou et al., 2015*). In a previous study, maximal inhibition of lidocaine on $I_h$ is ~50% (*Meng et al., 2011*). Therefore, adding more potent HCN channel modulators (e.g., ZD7288, clonidine and dexmedetomidine) to local anesthetics have been found to extend the durations of lidocaine and ropivacaine in vivo (*Brummett et al., 2011*; *Harris & Constanti, 1995*; *Kroin et al., 2004*).

Capsazepine (CPZ) is a competitive inhibitor of capsaicin activation of transient receptor potential vanilloid-1 (TRPV-1) channel and has been reported as an emerging class of novel analgesic (*Menendez et al., 2006*). TRPV-1 channel is mainly expressed on nociceptor and involved in sensory processing (*Cui et al., 2006*). TRPV-1 channel can be activated by high temperature, acid, capsaicin and other chemicals (*Szallasi et al., 2007*), by which to trigger action potentials and release of neuropeptides and/or transmitters (*Cui et al., 2006*; *Peters et al., 2010*). Therefore, TRPV-1 channel blocker might be potent in the therapy of pain. CPZ has also been found to inhibit HCN channel currents ($I_h$) (*Zuo et al., 2013*).

In the present study, we hypothesized that CPZ, a potential adjuvant for local anesthetics, might extend the durations of lidocaine by modulation of HCN channels. This study was designed to test this hypothesis.

## METHODS

### Animals

Following the ARRIVE Guidelines and with the approval of the Institutional Animal Experimental Ethics Committee of Sichuan University (Chengdu, Sichuan, China, protocol 2015014A), a total of 63 adult male C57BL/6J mice (20–30 g) for experiments of sciatic nerve block in vivo and in vitro and seven C57BL/6J mice at age of postnatal 8 days for experiment of isolated dorsal root ganglion (DRG) neurons were included. All the study mice were housed in standard condition with free access to food and water. All the mice were placed in the experimental environment for 1–2 h per day for consecutive 3 days before the formal experiments. The observers of the behavioral test were blinded to the group assignment of the study mice.

### Chemicals

Lidocaine solution was diluted from 2% (w/v) lidocaine (Shanghai Fortune Zhaohui Pharmaceutical Co., Ltd., Shanghai, China) with normal saline to the final concentration of 1% (w/v, ~35 mM). ZD7288 (Tocris Bioscience, Bristol, UK), the non-selective HCN channel blocker, and forskolin (Sigma-Aldrich, Co., St. Louis, MO, USA), the activator of adenylyl cyclase, were dissolved with normal saline. CPZ (Sigma-Aldrich, Co., St. Louis, MO, USA) was dissolved with Dimethyl sulfoxide (DMSO) (Sigma-Aldrich, Co., St. Louis, MO, USA) to a 2.5 mg/mL stock solution and then diluted with normal saline (for animal study in vivo) or bath solution (for electrophysiological recordings) to the final concentrations. For electrophysiological recordings, the final concentration of DMSO was 0.05% (v/v), which produces minimal effect in electrophysiological recordings (*Liu et al., 2016*; *Wang et al., 2016*).

### Patch clamping recordings

The vectors of pcDNA3-HE3 with mHCN1 (Mouse HCN1) and mHCN2 (Mouse HCN2) were expressed in human embryonic kidney 293 (HEK 293) cells as previously described (*Gill et al., 2004*). HEK 293 cells were cultured under standard procedures (*Meng et al., 2011*). The mHCN channel constructs were co-transfected with a green fluorescent protein plasmid (pEGFP; Clontech Laboratories, Mountain View, CA, USA) by Lipofectamine 2000 reagent (Invitrogen, Carlsbad, CA, USA). Recordings were performed 36–48 h after transfection. Whole-cell patch clamping recordings were performed at room temperature (~24–25 °C). Patch pipettes (with resistance of three to five MΩ) were made from the fire-polished borosilicate glass capillaries (Sutter Instrument Co., Novato, CA, USA). $I_h$ were sampled using an Axon 200B amplifier, digitized via a Digidata 1440A interface. Standard bath solution contained (mM) 118 NaCl, 25 KCl, two $MgCl_2$, two $CaCl_2$, 10 HEPES, and 10 glucose, pH = 7.4 adjusted by NaOH. Bath solution was continuously perfused at a speed of ~2 mL/min. The pipette solution contained (mM)

120 KCH$_4$SO$_3$, four NaCl, one MgCl$_2$, 0.5 CaCl$_2$, 10 HEPES, 10 EGTA, three Mg-ATP and 0.3 GTP-Tris, pH = 7.3 adjusted by KOH (*Meng et al., 2011*).

Hyperpolarization-activated currents ($I_h$) were evoked by hyperpolarized pulses (4 s) from −40 to −130 mV with an interval of −10 mV (*Gill et al., 2004*). Amplitudes of steady-state currents were produced at the end of hyperpolarizing voltage steps as tail currents, which used for the analysis of voltage dependence of channel gating. Tail currents were acquired at a fixed potential of −90 mV immediately after hyperpolarized pulses and were normalized and fitted with Boltzmann curves as: $G/G_{max} = 1/(1 + e(V_{1/2a} − V)/k)$, where $G/G_{max}$ = normalized conductance; $G_{max}$ = maximum conductance; $V_{1/2a}$ = voltage of half-maximum activation; $k$ = slope factor. Concentration-response of CPZ on the $I_h$ was fitted to the Hill equation: $Y = 1/(1 + 10(logIC_{50} − X) * h)$, where $Y$ = inhibitory ratio; $X$ = concentration of CPZ; $h$ = Hill coefficient.

Acute isolated DRG neurons were prepared for recordings of sodium channel currents ($I_{Na}$) and $I_h$. Briefly, mice were anesthetized by ketamine/xylazine (40/15 mg/kg). Then the DRGs of mice were removed and digested respectively in 1% papain for 45 min and in 1% collagenase for 30 min. DRG neurons were recorded after 2-h cell attachment to the coverslip. $I_h$ was evoked by hyperpolarized pulse (4 s) to −120 mV with the same solutions used for recordings on HEK 293 cells. To test the effect of intracellular cAMP, 200 μM cAMP was added to the patch pipette. $I_{Na}$ was recorded by depolarized pulse (20 ms) to 10 mV. Bath solution contained (in mM): 125 NaCl, 25 NaHCO$_3$, 1.25 NaH$_2$PO$_4$, 2.5 KCl, one MgCl$_2$, two CaCl$_2$, 15 glucose, 25 TEA-Cl. Internal solution contained (in mM): 110 CsF, nine NaCl, 1.8 MgCl$_2$, four Mg-ATP, 0.3 Na-GTP, 0.09 EGTA, 0.018 CaCl$_2$, nine HEPES, 10 TEA-Cl. CsOH was used to adjust to pH = 7.35 (290–310 mOsm).

### Sciatic nerve block model

Anesthetized with 2% isoflurane (Abbott Pharmaceutical Co. Ltd., Shanghai, China), total volume of 50 μL study solution was injected into the popliteal fossa of the mice and the needle was removed 5 s after injection. The injected site was then pressed for at least 5 s to prevent exudation of study solution. A total of 56 mice were randomly divided into seven groups ($n$ = 8/group): respectively receiving lidocaine alone, CPZ alone, ZD7288 (HCN channel blocker) alone, CPZ + lidocaine, ZD7288 + lidocaine, ZD7288 + CPZ + lidocaine, forskolin (an activator of adenylyl cyclase) + CPZ + lidocaine. The concentrations of lidocaine, ZD7288, CPZ and forskolin were 1% (w/v, ~35 mM), five mM, 53 and 153 μM, respectively (*Kroin et al., 2004*; *Zuo et al., 2013*). For the groups that received two or three drugs, ZD7288 and/or forskolin were injected 20 min before lidocaine and CPZ was injected 10 min before lidocaine (*Ando et al., 2004*; *Kroin et al., 2004*).

After the mice recovered to consciousness (~5 min after injection), a 25-G needle (Terumo Medical Corp., Tokyo, Japan) tied to a plastic fiber (~3 g equivalent to the von-frey fiber) was used to evaluate noxious sensory function of the injected limb. Noxious sensory block was defined as no purposeful aversive movements to the pinprick stimulus. Von-frey filaments (37450-277; Ugo Basile, Comerio, Italy) was used to test tactile

sensation of the injected limb and tactile sensory block was defined as no purposeful reactions to increased von-frey fibers more than two g compared to baselines. The specific site for application of pinprick and von-frey fiber was lateral plantar surfaces innervated by sciatic nerve. Thermal sensory function was determined by heat stimulus (Plantar test 37370; Ugo Basile, intensity of infrared radiation set at 60) applied to the injected limb. Thermal sensory block was defined as the withdraw latency of the mice increased more than 50% of the baseline latency (*Becker & Reed, 2012*). Cut-off time of thermal stimulus was 10 s to avoid injury. Motor function of mice was evaluated by the ability of hanging upside down with the injected paws (*Yamada et al., 2016*). Before injection, baselines of each function were measured. Because TRPV-1 channel blockers have been reported to induce dysregulation of body temperature (*Yang et al., 2015*), rectal temperature of the study mice was observed. For all the studied mice, rectal temperatures were normal (36–38 °C) during the regional anesthesia. All the mice completely recovered from regional anesthesia without any adverse events.

## Compound action potential recording on isolated sciatic nerves

Adult C57BL/6J mice were used. Bilateral sciatic nerves were obtained from the lumbar plexus to the knee. Then the sciatic nerves were put into a Ringer's solution (115.5 mM NaCl, two mM KCl, 1.8 mM CaCl$_2$, 1.3 mM Na$_2$HPO$_4$ and 0.7 mM NaH$_2$PO$_4$, pH = 7.35) for 20 min. BL-420N biological signal acquisition and analysis system (Techman Software Co. LTD, Chengdu, China) was used. Baseline values of compound action potentials (CAPs) were recorded every 5 min until 20 min to obtain a stable baseline. Then, the CAPs of each group were obtained after the sciatic nerve incubating into the drug solutions for 5 min. The parameters of the system were set as: voltage = one V; frequency = 100 Hz; duration of rectangular pulse = 0.1 ms (*Li et al., 2010*).

## Statistical analysis

SPSS 22.0 (SPSS Inc., Chicago, IL, USA) was used to analyze the data. The sample size ($n$ = 8/group) of the animal study in vivo was based on our preliminary test ($n$ = 4, not included in formal data). The duration of lidocaine for noxious sensory block was compared between the groups of lidocaine alone and lidocaine + CPZ (20.5 vs. 30.5 min). Then the sample size was calculated as 5.0 ($\alpha$ = 0.05; $\beta$ = 0.20). Duration data were calculated by averaging the duration of each mouse in the same group and expressed as mean ± SEM. The homogeneity of variance tests indicated that the duration data were normaly distributed. One-way analysis of variance (ANOVA) followed by post hoc of Bonferroni was applied to compare the durations between groups. Kaplan–Meier followed by *pair-wise* over strata of Log-rank was applied to compare the recovery curves of motor function block and noxious sensory block. For the results of electrophysiological recordings, all the data were presented as mean ± SEM. Voltage-dependent activation curves of $I_h$ were fit to the Boltzmann equation. Two-way ANOVA or student's *t*-test was applied to compare $I_h$ and channel gating as indicated. In all cases, $P < 0.05$ was considered as statistically significant.

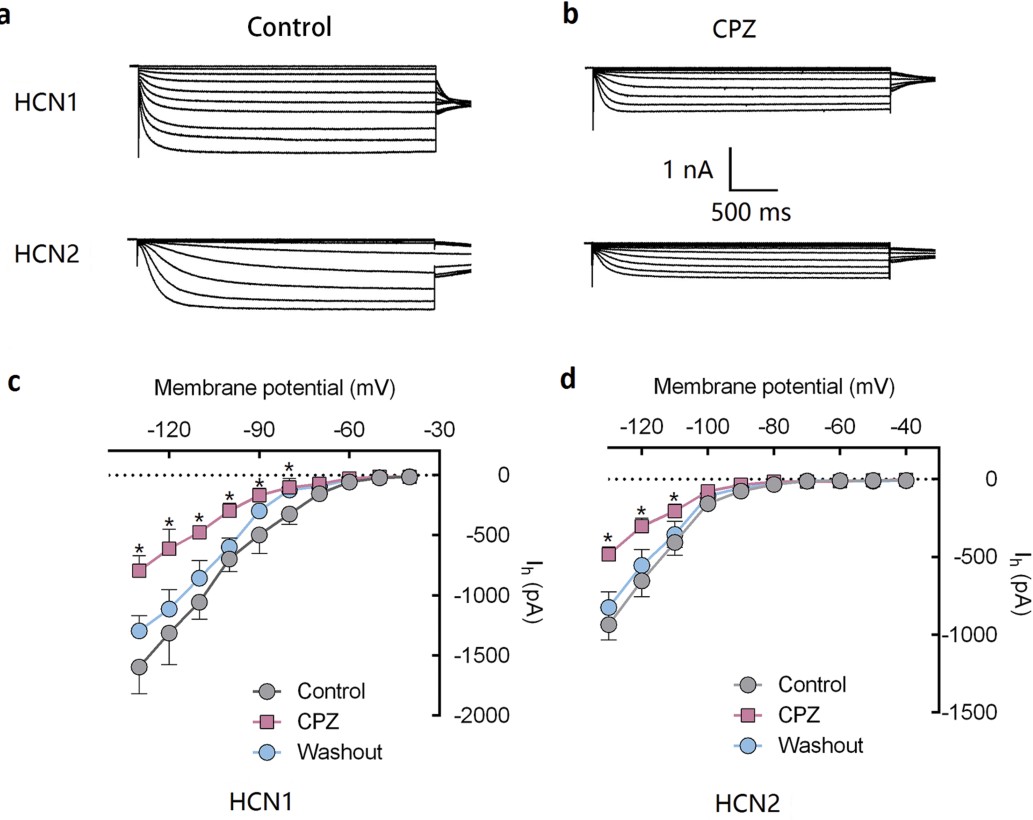

**Figure 1 CPZ inhibited HCN channel currents expressed on HEK 293 cells.** (A and B) CPZ inhibited HCN1 and HCN2 channel currents. (C and D) Effect of CPZ on I–V curves of HCN1 and HCN2 channels. Hyperpolarizing voltage steps from −40 to −130 mV evoked the currents of mHCN1 and mHCN2 channels. CPZ at 10 μM depressed the currents. Voltage steps were followed by a step to −90 mV for tail current analysis. CPZ effectively shifted voltage-dependent activation of HCN1 and HCN2 channels in a hyperpolarized direction. CPZ, capsazepine; *$P < 0.05$ vs. control group. Sample size = 5–6.

## RESULTS

### CPZ inhibited transfected mHCN channel currents and hyperpolarized voltage-dependent activation

Capsazepine at concentration of 10 μM potently inhibited I–V curves of both mHCN1 and mHCN2 channels (Fig. 1). Median inhibitory concentration (IC$_{50}$) of CPZ on peak $I_h$ were 8.2 ± 0.3 and 9.6 ± 0.4 μM, respectively for mHCN1 and mHCN2 ($P > 0.05$, Fig. 2). CPZ at concentration of 10 μM significantly caused a hyperpolarizing shift in voltage-dependence of channel activation for both mHCN1 (V$_{1/2a}$ of −98.4 ± 3.8 vs. −88.6 ± 3.6 mV, $P < 0.01$, Fig. 3) and mHCN2 (V$_{1/2a}$ of −102.7 ± 13.5 mV vs. −114.6 ± 12.9 mV, $P < 0.01$, Fig. 3).

### CPZ prolonged duration of lidocaine in sciatic nerve block model in vivo

ZD7288 alone (five mM) did not produce any sciatic nerve block in vivo. CPZ alone (53 μM) produced thermal sensory block with duration of 24.6 ± 2.7 min (Table 1).

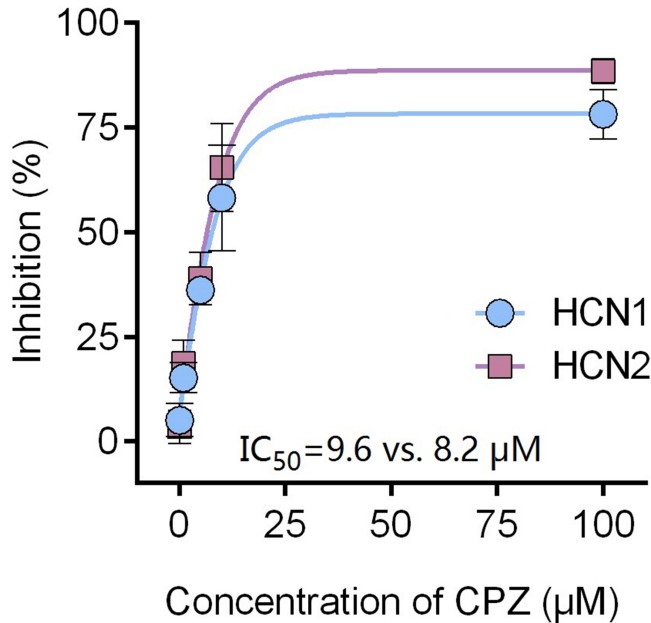

**Figure 2 CPZ concentration-dependently inhibited HCN1 (•) and HCN2 (■) channel currents.** The inhibitory potency of CPZ on HCN channel currents is similar between HCN1 and HCN2 channels, with $IC_{50}$ of 9.6 ± 0.4 and 8.2 ± 0.3 µM for HCN1 and HCN2, respectively. CPZ, capsazepine. Sample size = 6.

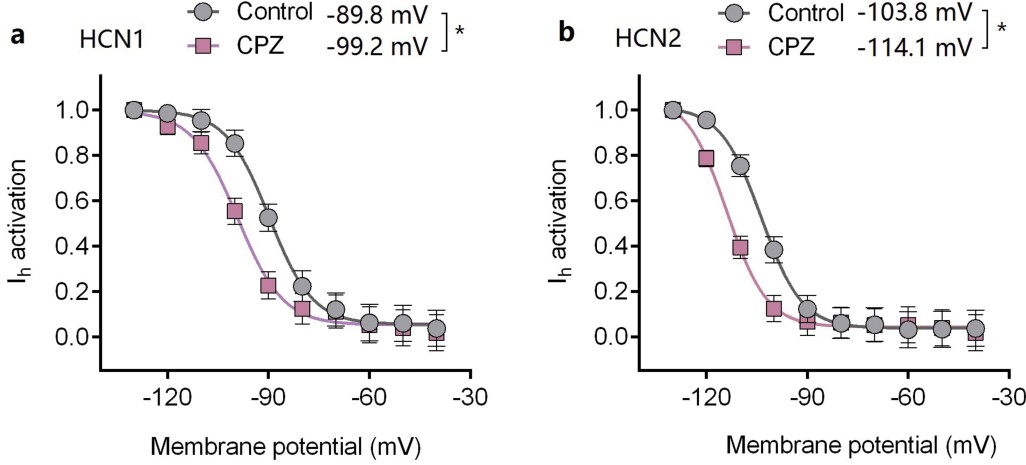

**Figure 3 Voltage-dependent activation curves of HCN1 and HCN2 channels.** (A) Voltage-dependent activation curve of HCN1 channel. (B) Voltage-dependent activation curve of HCN2 channel. CPZ significantly hyperpolarized the voltage-dependent activation of HCN1 and HCN2 channels. CPZ (10 µM) caused a hyperpolarizing shift in voltage-dependence of $V_{1/2act}$ from −89.8 ± 2.6 to −99.2 ± 3.6 mV (*$P < 0.01$) and from −103.8 ± 2.9 to −114.1 ± 4.8 mV (*$P < 0.01$), respectively for HCN1 and HCN2 channel. The voltage-dependent activation curves were based on tail currents. CPZ, capsazepine. Sample size = 7.

Compared with lidocaine alone, adding ZD7288 prolonged durations of lidocaine including noxious sensory block (34.3 ± 5.2 vs. 20.3 ± 1.7 min, $P < 0.001$, Fig. 4A), tactile sensory block (25.0 ± 4.3 vs. 20.0 ± 3.7 min, $P = 0.036$, Fig. 4B), thermal sensory block

**Table 1  Durations of sciatic nerve block for each group.**

| Groups | Durations (min) | | | |
|---|---|---|---|---|
| | Pin-prick | Von-frey | Thermal | Motor |
| ZD7288 | N/A | N/A | N/A | N/A |
| CPZ | N/A | N/A | $24.6 \pm 2.7$ | N/A |
| Lido | $20.3 \pm 1.7$ | $20.0 \pm 3.7$ | $26.8 \pm 5.5$ | $20.9 \pm 4.2$ |
| ZD7288 + Lido | $34.3 \pm 5.2^{*}$ | $25.0 \pm 4.3^{*}$ | $33.4 \pm 4.5^{*}$ | $27.9 \pm 4.3^{*}$ |
| CPZ + Lido | $35.1 \pm 3.3^{*}$ | $25.5 \pm 4.4^{*}$ | $39.6 \pm 6.6^{*\dagger}$ | $28.6 \pm 4.1^{*}$ |
| Forskolin + CPZ + Lido | $18.9 \pm 3.0^{\#}$ | $18.5 \pm 5.0^{\#}$ | $24.8 \pm 5.2^{\#}$ | $19.5 \pm 3.0^{\#}$ |
| ZD7288 + CPZ + Lido | $32.5 \pm 7.9$ | N/D | N/D | N/D |

Notes:
Data were expressed as mean $\pm$ SEM ($n$ = 8/group).
[*] $P < 0.05$ vs. Lido group.
[†] $P < 0.05$ vs. ZD7288 + Lido group.
[#] $P < 0.05$ vs. CPZ + Lido group.
Lido, 1% lidocaine; CPZ, capsazepine; N/A, no action; N/D, not determined.

($33.4 \pm 4.5$ vs. $26.8 \pm 5.5$ min, $P = 0.027$, Fig. 4C) and motor function block ($27.9 \pm 4.3$ vs. $20.9 \pm 4.2$ min, $P = 0.008$, Fig. 4D). Compared with ZD7288 + lidocaine group, CPZ similarly extended the durations of lidocaine including noxious sensory block, tactile sensory block and motor block. For noxious sensory block, CPZ prolonged the duration from $20.3 \pm 1.7$ to $35.1 \pm 3.3$ min ($P < 0.001$ vs. lidocaine alone, Fig. 4A). For tactile sensory block, CPZ prolonged the duration from $20.0 \pm 3.7$ to $25.5 \pm 4.4$ min ($P = 0.025$ vs. lidocaine alone, Fig. 4B). For motor function block, CPZ prolonged the duration from $20.9 \pm 4.2$ to $28.6 \pm 4.1$ min ($P = 0.003$ vs. lidocaine alone, Fig. 4D). For thermal sensory block, CPZ + lidocaine group was significantly longer than that of ZD7288 + lidocaine group ($39.6 \pm 6.6$ vs. $33.4 \pm 4.5$ min, $P = 0.017$, Fig. 4C). Adding ZD7288 and CPZ to lidocaine (ZD7288 + CPZ + Lido group) did not further prolong the duration of lidocaine for noxious sensory block compared to ZD7288 + Lido group or CPZ + Lido group (Fig. 4A). Of note, the durations of tactile sensory block, thermal sensory block and motor block in ZD7288 + CPZ + Lido group were not determined because slight sedation was found in some mice, which might affect the accuracy of the results. The pinprick stimulus (puncturing the skin by a 25-G needle) was much stronger than other stimulus like von-frey and in the manner of "all or none" compared with other quantitative measurements (e.g., von-frey and heat). No redness or swelling was found in the test limb during the whole experiment. All mice recovered completely after regional anesthesia. The time courses of all the regional anesthetic effects were shown in Fig. 5.

## Pre-injection of forskolin reversed the prolongation of CPZ in regional anesthesia

Pre-injection of forskolin (153 µM) diminished the prolongation of CPZ on durations of lidocaine (Figs. 4 and 5). The anesthetic durations were shorter in Forskolin + CPZ + Lido group than that in CPZ + Lido group including noxious sensory block ($35.1 \pm 3.3$ vs. $18.9 \pm 3.0$ min, $P < 0.001$, Fig. 4A), tactile sensory block ($25.5 \pm 4.4$ vs. $18.5 \pm 5.0$ min,

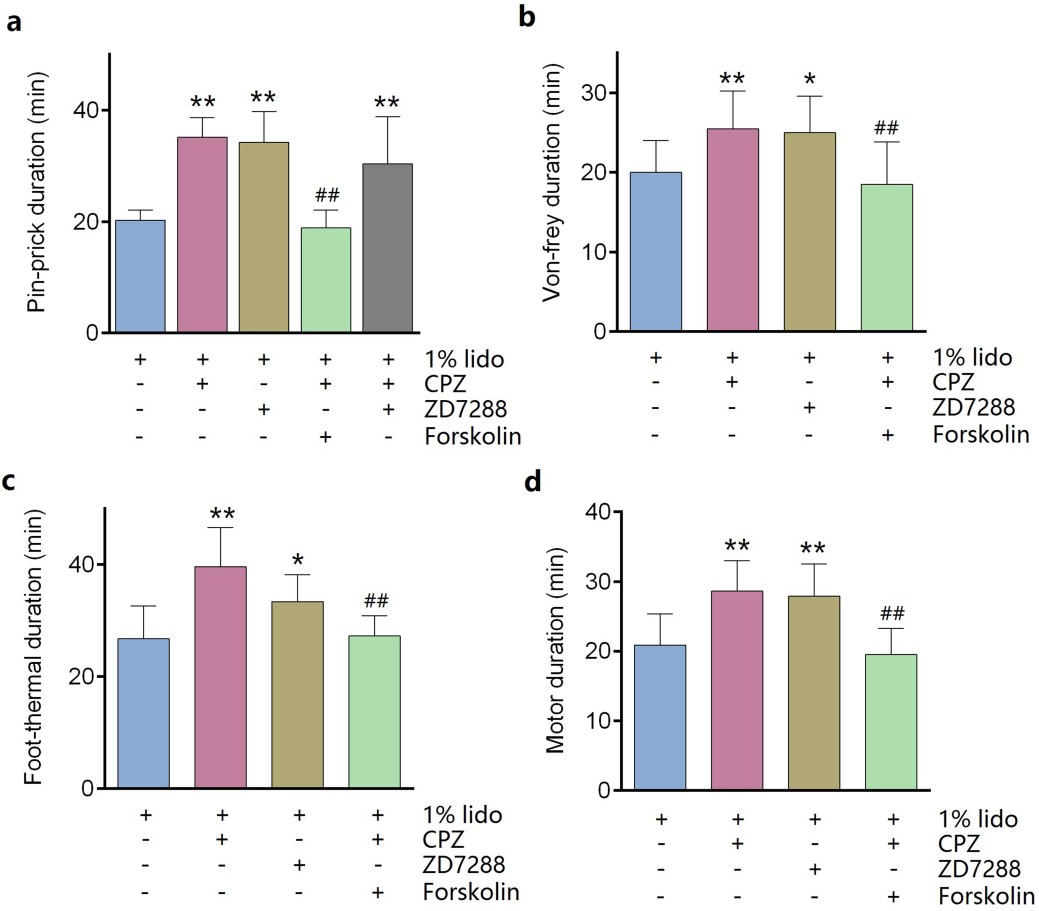

**Figure 4 Durations of sciatic nerve block produced by different drug combinations.** (A) Pin-prick test; (B) Von-frey test; (C): Foot-thermal test; (D) Motor function evaluation. Regional anesthetic durations of CPZ + Lido group and ZD7288 + Lido group were significantly longer than lido alone group, while forskolin reversed the prolongation by CPZ. In pin-prick test (A), ZD7288 + CPZ + Lido group did not produce any longer block than neither CPZ + Lido group nor ZD7288 + Lido group. For foot-thermal test (C), block duration of CPZ + Lido group was significantly longer than ZD7288 + Lido group. CPZ, capsazepine; Lido, lidocaine; $^*P < 0.05$ vs. Lido alone group; $^{**}P < 0.01$ vs. Lido alone group; $^{##}P < 0.01$ vs. CPZ + Lido group. Sample size = 8.

$P = 0.015$, Fig. 4B), thermal sensory blockade ($39.6 \pm 6.6$ vs. $24.8 \pm 5.2$ min, $P < 0.001$, Fig. 4C) and motor blockade ($28.6 \pm 4.1$ vs. $19.5 \pm 3.0$ min, $P = 0.001$, Fig. 4D). All the data were shown in Table 1. Of note, pre-injection of forskolin was prone to shorten the durations of lidocaine compared with lidocaine alone group, indicating the contribution of HCN channels in the effects of lidocaine (Figs. 5A, 5B and 5D).

## cAMP reversed the inhibitory effect of CPZ on $I_h$ in DRG neurons

The average current density of $I_h$ was $2.3 \pm 0.4$ pA/pF in control condition, while intracellular cAMP at 200 μM significantly increased the current density of $I_h$ to $7.6 \pm 2.3$ pA/pF ($P < 0.05$, Fig. 6B). CPZ at concentration of 10 μM inhibited $I_h$ density to $1.9 \pm 0.7$ pA/pF. Intracellular cAMP at 200 μM reversed $I_h$ density to $7.1 \pm 1.6$ pA/pF even with perfusion of 10 μM CPZ ($P < 0.05$, Fig. 6D).

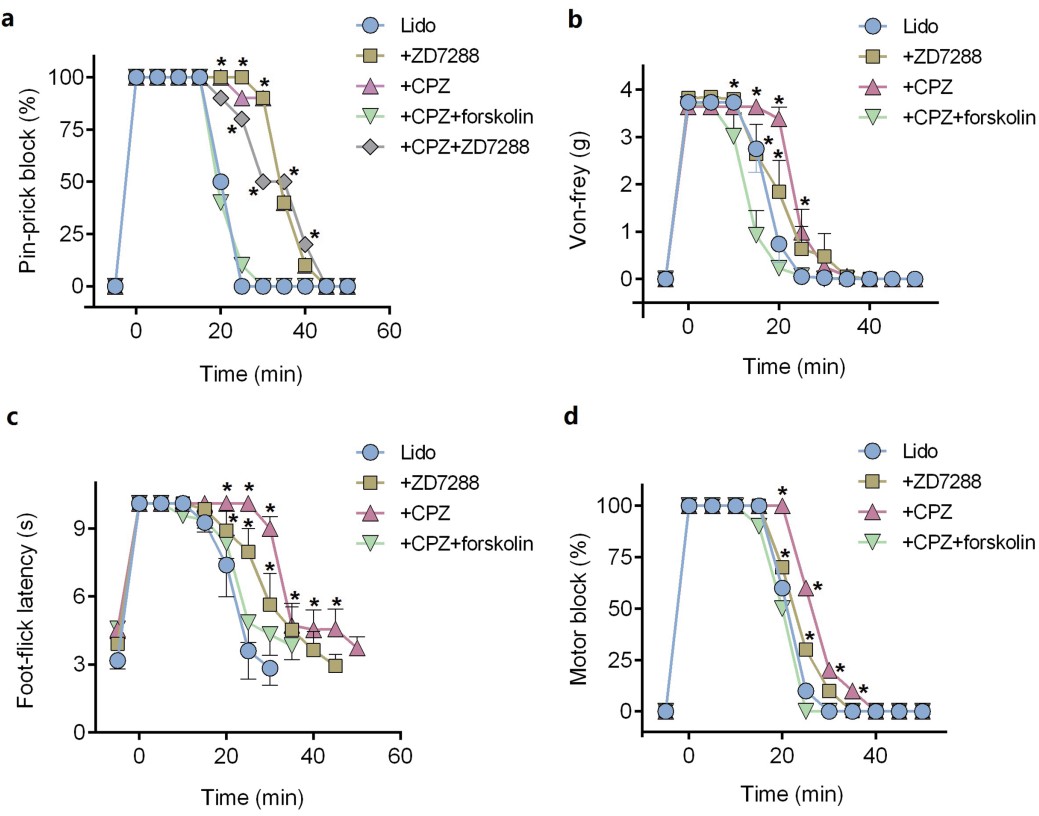

**Figure 5 Onset and recovery of regional anesthetic effects by different tests.** (A) Pin-prick test; (B) Von-frey test; (C) Foot-thermal test; (D) Motor function evaluation. For pin-prick test (A) and motor function evaluation (D), the time-dependent curves indicate block percentage at each time point. For von-frey test (B) and foot-thermal test (C), the time-dependent curves were calculated by averaging the results at each time point. CPZ, capsazepine; Lido, lidocaine; *$P < 0.05$ vs. Lido alone group. Sample size = 8.

## CPZ enhanced inhibition of lidocaine on action potential in sciatic nerves

Capsazepine at concentration of 10 μM did not inhibit $I_{Na}$ in isolated mice DRG neurons (Fig. 7A), while 30 μM CPZ significantly inhibited the current density of $I_{Na}$ from 105.9 ± 23.3 to 68.2 ± 20.3 pA/pF ($P < 0.05$, Fig. 7C). However, neither 10 nor 30 μM CPZ inhibited the CAP amplitudes in isolated sciatic nerves compared to baseline (Figs. 7E–7H). The CAP amplitudes were inhibited by 43.3% ± 1.5% after incubation in 1% lidocaine for 5 min. CPZ dose-dependently enhanced the inhibitory effect of 1% lidocaine on the CAP. For 1% lidocaine + 10 μM CPZ group and/or 1% lidocaine + 30 μM CPZ group, the CAP amplitudes were decreased to 20.5% ± 3.0% and/or 12.6% ± 0.8% compared to baseline, respectively ($P < 0.01$ vs. 1% lidocaine, Fig. 7I).

## DISCUSSION

Long-lasting local anesthetics are demanded in clinic especially for post-operative analgesia and treatment of chronic pain. To prolong duration of local anesthetics and reduce their
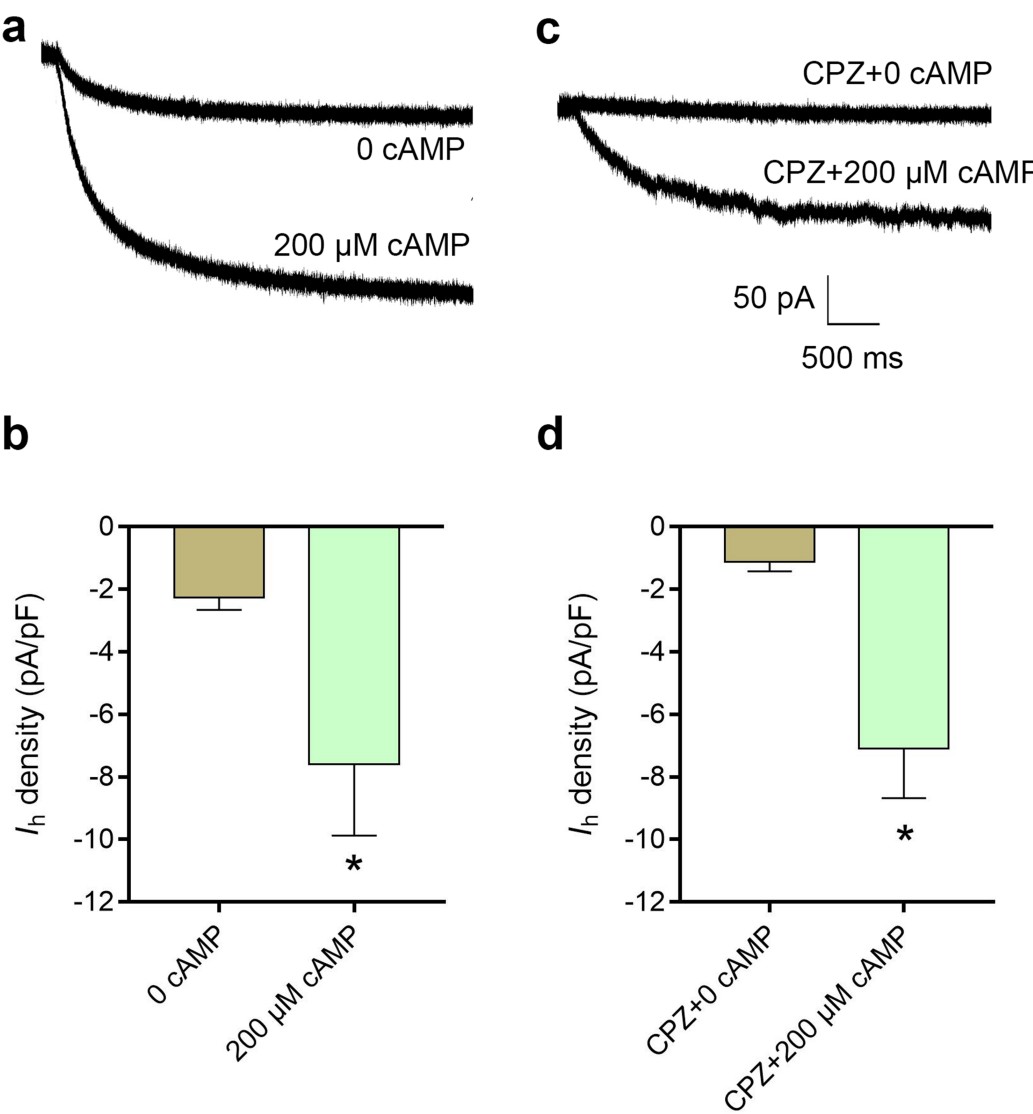

**Figure 6 Increased intracellular cAMP reversed the inhibitory effect of CPZ on HCN channel currents ($I_h$) in mice DRG neurons.** (A) Representative traces of the $I_h$ activated by 200 µM cAMP. (B) Current density of $I_h$ in DRG neurons of mice significantly increased from 2.3 ± 0.4 pA/pF to 7.6 ± 2.3 pA/pF when cAMP (200 µM) was introduced to the cytoplasm from the patch pipette. (C) Representative $I_h$ traces when CPZ (10 µM) added with and without cAMP. (D) CPZ inhibited $I_h$ density in mice DRG neurons to 1.9 ± 0.7 pA/pF, while cAMP reversed to 7.1 ± 1.6 pA/pF. CPZ, capsazepine; DRG, dorsal root ganglion; $^*P < 0.05$. Sample size = 8–12.     

toxicities, many adjuvants have been added to local anesthetics (*Yilmaz-Rastoder et al., 2012*). The adjuvants for local anesthetics mainly include epinephrine, ketamine, opioids and NSAIDs (*Bailard, Ortiz & Flores, 2014*). However, the enhancement of most adjuvants are unsatisfied and some adjuvants can even induce unacceptable side effects themselves, such as poor selectivity, undesired systematic sedation or thermal irritation (*Brummett & Williams, 2011*). In the present study, CPZ significantly extended the durations of lidocaine for sciatic nerve block without obvious complications. Therefore, CPZ may be a potential adjuvant for local anesthetics.

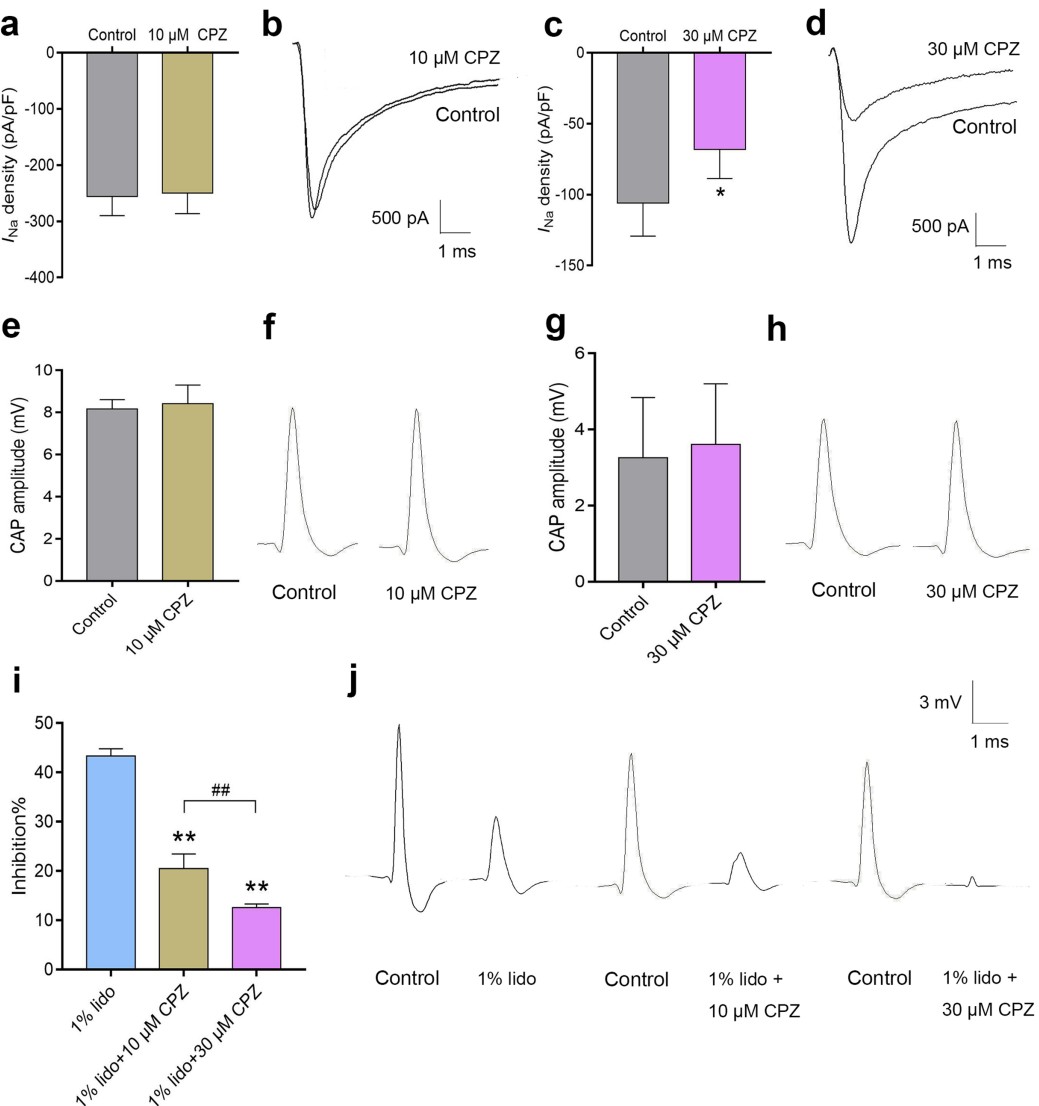

**Figure 7 The effects of CPZ on voltage-gated sodium channel current ($I_{Na}$) in DRG neurons and compound action potentials (CAP) amplitudes in sciatic nerves of mice.** (A–D) CPZ at 10 µM showed no influence on $I_{Na}$ (A), while 30 µM CPZ significantly inhibited current density of $I_{Na}$ (C). The rightmost traces (B and D) show samples of $I_{Na}$ after perfusion of 10 and 30 µM CPZ, respectively. (E–H) CPZ at 10 or 30 µM did not inhibit CAP amplitudes in mice sciatic nerves. Samples (F and H) are shown in the right. (I) CPZ at 10 µM and/or 30 µM enhanced the inhibitory effect of 1% lidocaine on CAP amplitudes. (J) Are the representative traces. CPZ, capsazepine; DRG, dorsal root ganglion; CAP, compound action potentials; [*]$P < 0.05$ vs. control group; [**]$P < 0.01$ vs. 1% lidocaine group; [##]$P < 0.01$ vs. 1% lidocaine + 10 µM CPZ group. Sample size = 3–5.

Capsazepine is a competitive inhibitor of capsaicin activation of TRPV-1 channel (*Yang et al., 2015*). At micromolar concentrations, CPZ can also block voltage-dependent potassium channel, calcium channel and sodium channel (*Kuenzi & Dale, 1996*; *Yamamura et al., 2004*; *Lundbaek et al., 2005*). In the present study, to test whether voltage-gated sodium channels are involved in the enhancement of CPZ on regional anesthesia of lidocaine in vivo, acute DRG neurons and isolated sciatic nerves were used

for recordings of $I_{Na}$ and CAP, respectively. CPZ alone at a concentration of 30 μM indeed inhibited $I_{Na}$ in DRG neurons. However, CPZ did not affect amplitudes and/or conductions of CAP in isolated sciatic nerves at concentration of 30 μM. A previous study indicates that inhibition of CPZ on $I_{Na}$ is possibly by altering lipid bilayer elasticity of the cellular membrane (*Lundbaek et al., 2005*). Therefore, the differential results of CPZ between voltage-gated $I_{Na}$ and action potential might result from the myelin sheath around nerves that preventing direct exposure of CPZ to the cellular membrane. In spite of no effect was found for CPZ alone on CAP, it can enhance the inhibitory effect of lidocaine on CAP amplitudes since the concentration of 10 μM. Consistent with the observation in vivo, CPZ alone did not produce typical sciatic nerve block, but prolonged the durations of lidocaine. Although CPZ may not inhibit action potential alone by inhibition to sodium channels, it cannot completely exclude the possible contribution of sodium channel block by CPZ for its enhancement in regional anesthesia. Even a small additional inhibition of sodium channel might prolong the effects of lidocaine.

Capsazepine can inhibit transfected human HCN channel currents (*Zuo et al., 2013*). HCN channel is widely expressed in the peripheral nervous system (*Acosta et al., 2012*; *Cao, Pang & Zhou, 2016*). HCN channels may contribute to the regional anesthetic effects of lidocaine in vivo, both for the sensory and motor function block (*Zhou et al., 2015*). By electrophysiological recordings, lidocaine was found to inhibit $I_h$ (*Meng et al., 2011*). However, lidocaine at clinical concentrations only inhibits $I_h$ by ∼30–50% (*Meng et al., 2011*). Thus potent HCN channel blocker might enhance the effects of local anesthetics (*Brummett et al., 2011*). HCN channel modulators (e.g., ZD7288, clonidine and dexmedetomidine) have been found to enhance regional anesthetic effects of lidocaine and/or ropivacaine in vivo (*Brummett et al., 2011*; *Harris & Constanti, 1995*; *Kroin et al., 2004*). Although CPZ can inhibit transfected human HCN channel currents (*Zuo et al., 2013*), it is unclear whether CPZ can enhance regional anesthesia in vivo. In the present study, CPZ inhibits transfected mice HCN channel currents without selectivity between HCN1 and HCN2. The $IC_{50}$ of CPZ measured in this study (∼10 μM) is similar to the potency on human $I_h$ (*Gill et al., 2004*; *Zuo et al., 2013*), indicating the enhancement of CPZ for regional anesthesia might be conserved between species.

ZD7288 and CPZ similarly prolong the durations of lidocaine. Adding both ZD7288 and CPZ to lidocaine did not further prolong the duration of noxious sensory block compared with ZD7288 + Lido or CPZ + Lido group. Therefore, enhancement of ZD7288 and CPZ in regional anesthesia might be both mediated through modulation of $I_h$. For DRG neurons, intracellular cAMP reversed the inhibitory effect of CPZ on $I_h$. The enhancement by CPZ for lidocaine in sciatic nerve block was eliminated by pre-injection of forskolin. Forskolin can increase the intracellular level of cAMP (*Brummett et al., 2011*) and is widely used to enhance $I_h$ (*Wainger et al., 2001*; *Kroin et al., 2004*). These results provide better evidence to support the hypothesis that modulation of CPZ on $I_h$ may contribute to its enhancement in regional anesthesia. Forskolin will also likely increase local blood flow and enhance drug clearance. However, the concentration of forskolin used in this study was 153 μM, which is lower than the blood concentration (∼400 μM) to influence the blood flow (*Wysham, Brotherton & Heistad, 1986*). Forskolin

reversed the prolongation of CPZ most potently in von-frey test than pin-prick test, foot-thermal test and motor function evaluation (Fig. 5); thus the selective effect of forskolin might not be explained by increasing the blood flow (no selectivity). Interestingly, the enhancement of cAMP between HCN1 and HCN2 is different (*Wainger et al., 2001*), which might contribute to the differential reversion by forskolin between different functions.

Capsazepine alone can produce thermal sensory block without lidocaine. The thermal sensory block duration in CPZ + lidocaine group is longer than that of ZD7288 + lidocaine group. The selective enhancement of CPZ for thermal sensory might result from the inhibition of TRPV-1 channels. TRPV-1 channel blocker is a potential treatment for various types of pain (*Walker et al., 2003*). However, because of the role of TRPV-1 channel as a temperature sensor, systemic administration of TRPV-1 channel blocker can induce serious hyperpyrexia (*Menendez et al., 2006*), which limits the clinical application of TRPV-1 channel blocker. The systemic dosage of CPZ to induce hyperpyrexia is unknown, but other TRPV-1 antagonists such as AMG517, GRC6211 and NGD8243 have been dropped out of clinical trials due to hyperpyrexia (*Pareek et al., 2007*). In this study, CPZ is perineurally injected and no obvious change was found for mice body temperature.

There are some limitations in the present study: Firstly, forskolin, an activator of adenylyl cyclase, is not a selective agonist of HCN channels. Forskolin can increase the level of cAMP, which is an enhancer of HCN channels (*Wainger et al., 2001*; *Kroin et al., 2004*). Secondly, ZD7288 and CPZ are non-selective blockers between HCN1 and HCN2 subtypes. Thus, we cannot evaluate the specific role of HCN1 and HCN2. Further studies with HCN selective blockers such as MEL57A and/or EC18 might shed light on this question (*Del Lungo et al., 2012*). In addition, the potential effect of CPZ on HCN4 (cardiac subtype) should be tested further to explore the potential influence of CPZ on the heart. Co-injection of vasoconstrictor and CPZ may not only prolong the durations of local anesthetics, but also reduce the systemic impact of CPZ in regional anesthesia.

## CONCLUSIONS

In summary, CPZ, a widely known competitive inhibitor of capsaicin activation of TRPV-1 channel, can inhibit $I_h$ and prolong durations of lidocaine in vivo. The enhancement of CPZ in regional anesthesia may be mediated by modulation of $I_h$. Therefore, CPZ is a potential regional anesthetic adjuvant in the clinic.

### Funding

This work was supported by the grant 81771456 (to Cheng Zhou) and 81571353 (to Jin Liu) from the National Natural Science Foundation of China. The funders had no role in the study design, data collection and analysis, decision to publish, or preparation of the manuscript.

## Grant Disclosures

The following grant information was disclosed by the authors:
National Natural Science Foundation of China: 81771456 (to Cheng Zhou) and 81571353 (to Jin Liu).
National Natural Science Foundation of China.

## Competing Interests

The authors declare that they have no competing interests.

## Author Contributions

- Wenling Zhao performed the experiments, analyzed the data, prepared figures and/or tables, authored or reviewed drafts of the paper, approved the final draft.
- Peng Liang performed the experiments, contributed reagents/materials/analysis tools.
- Jin Liu contributed reagents/materials/analysis tools.
- Huan Li analyzed the data.
- Daqing Liao analyzed the data.
- Xiangdong Chen analyzed the data, authored or reviewed drafts of the paper.
- Qian Li conceived and designed the experiments, approved the final draft.
- Cheng Zhou conceived and designed the experiments, prepared figures and/or tables, approved the final draft.

## Animal Ethics

The following information was supplied relating to ethical approvals (i.e., approving body and any reference numbers):

The study was approved by the Institutional Animal Experimental Ethics Committee of Sichuan University (2015014A).

## Data Availability

The raw measurements are available as Supplemental Files.

## Supplemental Information

Supplemental information for this article can be found online at http://dx.doi.org/10.7717/peerj.7111#supplemental-information.

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
