# Peer review of "Capsazepine prolongation of the duration of lidocaine block of sensory transmission in mice may be mediated by modulation of HCN channel currents"

_PeerJ, doi:10.7717/peerj.7111_

## Round 0.1 · original submission · Major Revisions

This study was fairly well received by the Reviewers, and the results are clear. However, the authors do need to temper their conclusions, and have the Ms corrected by a native English speaker. More experiments would likely add to the impact of the paper, but they are not strictly necessary (if conclusions are suitably moderated). Clarification of the points raised by the Reviewers, and below, are necessary.

Fundamentally, they have not demonstrated that capsazepine is inhibiting IH in vivo to produce enhanced sensory nerve block. The authors should fully consider the effects of capsazepine on Na channels themselves, and also think about the effects of forskolin on other conductances likely to be important in sensory nerves – including TRPV1 and NaV1.6, to name just a couple. FSK will also likely increase local blood flow and enhance drug clearance – any of/all of these factors could contribute to the occlusion of the effect of capsazepine on nerve conduction. The only way to know for sure whether capsazepine is acting strongly at IH in vivo is to do experiments in HCN knockout animals (not that I am suggesting this is necessary in this study). The role of targets other than TRPV1 could be tested with capsazepine ± a more selective TRPV1 antagonist – again, the drugs are out there.

There is also no particular reason why shifting IH activation to more positive potentials with FSK would occlude block by capsazepine – this could be tested in vitro, and probably should be. The authors should also probably discuss what other channels capsazepine might modulate at these concentrations – there is data out there.

The authors should clarify whether the concentration of DMSO was the same for all experiments in vitro, and it is essential to report the shifts in HCN activation produced by DMSO alone (i.e., by time). Illustrations of time courses are always helpful. It would also be nice to know if the effects of capsazepine on HCN channels reversed on wash.

Clonidine and dexmetomidine are primarily alpha2 receptor agonists … this may inhibit IH via modulation of cAMP levels, but direct inhibition of the channel is unlikely to be their major mechanism of action. The molecular actions of drugs should be introduced at first mention – introduce forskolin as an activator of adenylyl cyclase, not an “IH activator”.

·

Basic reporting

Although the native English reader is able to understand the context of this study, the language used needs some revision.

Experimental design

This is a reasonably well designed study with standard electrophysiological approaches employed for the HCN channel experiments in transfected cells. Behavioural studies also follow standard approaches as do the experimental analyses.

Validity of the findings

The key finding, about which all behavioural interpretation is based, is that this study replicates the results of an earlier (2013) study demonstrating that capsazepine blocks HCN channels. The 2013 study was interesting by did not really show the mechanism and focussed on the two HCN channel isotypes (HCN2 and HCN4) that are modulated strongly by cAMP, thus allowing for multiple potential mechanisms. Thus it remains unclear what the mechanisms may be. Since the current study is based on this 2013 study, it is a bit of a reach to suggest in the title and throughout the manuscript that capsazepine is exerting its effects through blockade of HCN channels.

Additional comments

This is an interesting study, with conclusions that are based a little too strongly around a mechanism (how capsazepine modulates HCN channels) that is yet to be definitively resolved. As addressed in the discussion, there is a great need for prolonging regional anaesthesia. If a pan HCN blocker is going to be incorporated into regional anaesthesia it will be important to maintain the current best practice of vasoconstriction around the injection sites to prevent the potential impact on the HCN4 channels in the sino atrial node?

Minor comments:
Ln 72: "...forskolin .... HCN channel enhancer". Although this is how the drug is used in the current study, this is not correct. forskolin stimulates adenylyl cyclase, increasing cAMP which modulates some (HCN2 and HCN4) HCN channels.

Lines 181-6: combinations of lidocaine with ZD and CPZ caused sedation in some animals thus rendering the results from some behavioural tests ambiguous. The authors have excluded all but the pin prick test- would the sedation not also potentially alter these results?

Pin prick test experiments: repeatedly puncturing the skin will cause a localised inflammatory reaction and thus alter the threshold for activating nociceptors (and other neurons).

Table 1: not clear what the comparison is for the ZD + CPS + lido group.

Reviewer 2 ·

Basic reporting

Very poorly written paper.
Literature references are inadequate as several statements have no or inappropriate references

The structure of the article is fine and the data are well presented.

Experimental design

No comments

Validity of the findings

No comment

Additional comments

In this study, Zhao et al examined whether the TRPV-1 channel antagonist, capsazepine (CPZ) could prolong duration of lidocaine action in mice using different approaches. Consistent with their hypothesis, they found that CPZ prolonged duration of lidocaine, and they concluded that CPZ exerts its action by inhibiting both HCN and TRPV-1 channels.


Overall, the study addressed, using different appropriate approaches, an important issue; the experiments seemed to be well conducted with the data being well presented. Unfortunately, however, the paper is poorly written which makes assessing the science in this paper a very hard task. Indeed, there are many arguments that are hard to follow and there are many redundant and irrelevant statements that often not supported by references.
For example, no reference is provided for the statement (Line 45) “ TRPV-1 channel is involved in sensory processing and mainly expressed on nociceptor” .
The followings are examples of sentences that are either do not make sense or hard to follow:
L32: For the distinct electrophysiological properties of HCN channels, it is the principal pacemaker for rhythmical regulator and determinant for resting membrane potential in nervous system and sinoatrial node cells (Benarroch, 2013; Biel et al., 2009)”.
L38: Interestingly, because the maximal inhibition of lidocaine on HCN channel currents is only ~50% (Meng et al., 2011), therefore more potent HCN channel blockers including ZD7288, clonidine, and dexmedetomidine have been found to extend the nerve block durations for lidocaine and ropivacaine (Brummett et al., 2011; …”
L62: All the study mice were housed in standard conditions
L63: Before formal experiments, mice were…..

---

## Round 0.2 · Major Revisions

The authors have responded to the comments of the editors and reviewers by including arguments that they feel strengthen their case that capsazepine block of IH is involved in enhancing the regional analgesia produced by lidocaine. Unfortunately, some of the papers they offer in support of their hypothesis are not accurately interpreted, and the authors have ignored some work which does not support their ideas. The experimental design the authors have used has provided evidence for other drugs acting via IH block to enhance regional anaesthesia, but simply following a well worn path is not enough to establish a mechanism of action, particularly when no recording of nerve potentials are made. At this stage, only further experiments are needed to support the suggestion that capsazepine may be acting at IH to affect sensory nerve function, in particular, the idea that FSK will occlude the effects of capsazepine must be established in vitro at least.

The Ms still needs the attention of a language editor, the Introduction in particular is stilted, and new text in the Results is also hard to make sense of. The new title does not make sense as written.

The authors repeatedly state that they have tempered their conclusions, yet the conclusion reads:

“CPZ can prolong durations of lidocaine in peripheral nerve block by modulation of HCN channel currents”,
Which is a pretty strong statement, and not supported by the data. This needs to be tempered with “may”.

The authors claim that capsazepine does not inhibit Na channels (the principle target of lidocaine), and they cite 2 papers in support (Kuenzi & Dale, BJP; Yamamura et al., JBC); however, neither paper actually assesses whether voltage gated Na channels are a target for capsazepine. This is at best careless, at worst disingenuous. More importantly, the authors fail to cite work by Lundbaek et al (Mol Pharm 68, 680) showing very clearly that capsazepine blocks voltage gated Na channels at concentrations used in this work (30 µM in the Lundbaek study, 50 µM in the present work). This is an important point, because while the authors use a dose of lidocaine that is approximately 2x the EC50 to block nerve activity, this does not mean that all the Na channels in the nerve are blocked (an interpretation stated several times in the Ms) – block of the sensory response means there are simply enough channels blocked to inhibit conduction, this does not mean 100% of channels. Addition of a drug which can by itself inhibit Na channels could easily prolong the effects of lidocaine by additive means. Any revised Ms needs to include an appropriate discussion of the possibility of NA channel block by capsazepine.

The authors make much of the fact that forskolin is an (indirect) activator of IH, though its effects on cAMP concentrations. Increasing cAMP promotes voltage-dependent activation of the channel. This is true, but elevating cAMP also potentiates Na currents, and affects many other channels. The authors state, without evidence, that the effects of cAMP on IH gating will reverse the effects of capsazepine. We don’t actually know this, and it cannot be assumed that one effect on gating will necessarily override another. The interaction of forskolin and capsazepine needs to be experimentally tested in any revised Ms, doing so would provide better evidence to support the authors’ hypothesis.

The authors cite work purporting to show that the alpha2 agonists clonidine and dexmedetomidine act through IH blockade to produce/prolong analgesia (Yang, EJP; Yang NeuroReport), as support for this putative mechanism of action of capsazepine. Some of these papers are clear, others not. Yang (EJP) shows only a small difference in the potency of clonidine in HCN knockout animals, and further that the clonidine analgesia is almost completely reversed by the adrenoceptor antagonist yohimbine – which actually provides evidence for modest HCN involvement if any. In the other paper (Yang, NeuroReport), it is unclear whether the involvement of alpha2 receptors in the effects of the alpha2 agonists are even tested. It is hard to draw a direct connection between an in vitro effect on a channel and complex in vivo measurements, particularly for antagonists, but simply asserting that something is happening does not make it so. Clonidine and dexmedetomidine are also agonists, not “blockers” of alpha2 receptors (line 271).

Discussion of the greater potency of capsazepine versus ZD7288 is spurious as no dose/response curves were performed for either drug in vivo.

Capsazepine is not a “competitive inhibitor of TRPV1”, it is a competitive inhibitor of capsaicin activation of TRPV1.

The authors were asked to include details of solvent concentrations in vitro, they replied with “<1% DMSO”, which was claimed to have no effect on channels. The authors need to report what the concentrations of solvent actually were, and whether they were the same for each condition/drug concentration.

---

## Round 0.3 · Minor Revisions

Thank you for constructively addressing the comments of the Reviewers in your response, particularly through your additional experiments. The only major issue is now the Title of the Ms, which does not make sense as written. Perhaps "Capsazepine prolongation of the duration of lidocaine block of sensory transmission may be mediated by modulation of HCN channel currents" (or something similar). There is still a little work to be done with the language of the Ms - just one example is line 221 "No red and swollen was found in the test limb during the whole experiment" ... redness, swelling, would be more appropriate.

---

## Round 0.4 · accepted · Accept

Thank you for your prompt attention to the title of your paper.